# Explainable Machine-Learning-Based Characterization of Abnormal Cortical Activities for Working Memory of Restless Legs Syndrome Patients

**DOI:** 10.3390/s22207792

**Published:** 2022-10-14

**Authors:** Minju Kim, Hyun Kim, Pukyeong Seo, Ki-Young Jung, Kyung Hwan Kim

**Affiliations:** 1Department of Biomedical Engineering, College of Health Science, Yonsei University, 1, Yeonsedae-gil, Heungeop-myeon, Wonju-si 26493, Korea; 2Department of Neurology, Seoul National University Hospital, Seoul National University College of Medicine, 101, Daehak-ro, Jongno-gu, Seoul 03080, Korea

**Keywords:** restless legs syndrome, working memory, event-related potential, explainable machine learning, convolutional neural network

## Abstract

Restless legs syndrome (RLS) is a sensorimotor disorder accompanied by a strong urge to move the legs and an unpleasant sensation in the legs, and is known to accompany prefrontal dysfunction. Here, we aimed to clarify the neural mechanism of working memory deficits associated with RLS using machine-learning-based analysis of single-trial neural activities. A convolutional neural network classifier was developed to discriminate the cortical activities between RLS patients and normal controls. A layer-wise relevance propagation was applied to the trained classifier in order to determine the critical nodes in the input layer for the output decision, i.e., the time/location of cortical activities discriminating RLS patients and normal controls during working memory tasks. Our method provided high classification accuracy (~94%) from single-trial event-related potentials, which are known to suffer from high inter-trial/inter-subject variation and low signal-to-noise ratio, after strict separation of training/test/validation data according to leave-one-subject-out cross-validation. The determined critical areas overlapped with the cortical substrates of working memory, and the neural activities in these areas were correlated with some significant clinical scores of RLS.

## 1. Introduction

Restless legs syndrome (RLS) is a sensorimotor disorder accompanied by a strong urge to move the legs and an unpleasant sensation in the legs [1,2]. Most patients with RLS complain of several sleep disorders, including insomnia and poor sleep quality. Prefrontal lobe function deterioration has been reported to be associated with RLS, for example, reduced test scores in the verbal fluency test and the Trail Making Test [3,4]. Several studies also reported working memory (WM) deficits in RLS patients [3,5,6], which also implies prefrontal region dysfunction considering the recognized role of this area in working memory [7,8].

Previously, we showed that the working memory dysfunction in RLS is accompanied by reduced P300 event-related potential (ERP) amplitude compared with normal controls during working memory retrieval, and the P300 amplitude reduction is correlated with the duration of RLS history [5]. In addition, the phase synchronization between the theta band activities of frontal and posterior areas was also significantly decreased in RLS patients [6]. These results seem to reflect the problems of attention allocation and stimulus evaluation associated with RLS. Thus, frontal lobe dysfunction has been consistently implied in RLS patients [3,4,5,6,9,10,11,12]. The dopaminergic pathway linking prefrontal cortex and ventral tegmental area is known to play an important role in frontal cognitive function [13]. This is in line with low iron in deep brain structures, which are common in RLS patients and known to inhibit dopaminergic function [10,11]. The neural mechanism of cognitive deficit in RLS including working memory has not been sufficiently elucidated in spite of several electrophysiological studies.

In conventional statistical analysis of ERP, a significant amount of information is lost as a result of averaging of single-trial activities for each individual subject. Another problem is that the statistical comparison of cortical activities at thousands of cortical points converted from scalp electroencephalogram (EEG) results in a heavy multiple comparison problem [14], thus it becomes difficult to find out information on the critical cortical points, which is essential to identify abnormal cortical functions in RLS patients. Our approach can avoid these problems of conventional statistical analysis by identifying the critical input nodes that most significantly affected the performance of the CNN classifier. These input nodes correspond to the cortical regions, in which the neural activity is substantially different between normal controls and RLS patients. Recent advances in machine learning, especially deep neural networks, have enabled improved analysis of multidimensional data such as high-density EEG, and are extensively adopted for pattern recognition and the estimation of neural information [15,16,17,18].

Structural magnetic resonance imaging (MRI) is not suitable to clarify the cortical regions that show functional differences between the RLS patient and normal controls. Our purpose was to identify the regions that may underlie impaired cognitive functions associated with the RLS, especially during the working memory task. Functional MRI may be applied to reconfirm our results, but, because of the low temporal resolution, it is not possible to extract the neural activity during the short time interval, which is critical for performing the task.

In this study, we aimed to reveal the neural mechanism of working memory deficits associated with RLS using machine-learning-based analysis of cortical activities derived from multichannel ERP recordings. A pattern classifier based on a convolutional neural network (CNN) was developed to discriminate the cortical activities between RLS patients and normal controls. An explainable machine learning approach was applied to the trained classifier to determine the critical nodes in the input layer for the output decision. These critical input nodes can be regarded to represent the time/location of cortical activities discriminating RLS patients and normal controls during working memory tasks. Our methods enabled highly accurate discrimination of RLS patients based on single-trial ERP. Considering the accuracy and generalization performances, this may provide the basis for a prescreening tool for the RLS.

Recent studies have applied deep neural network for the EEG classification and have reported on important EEG features for the decision of the neural network [18,19,20,21]. Those studies used two-dimensional (2D) images of time–frequency representation or scalp topography as the inputs to the deep neural network. Here, we applied a deep neural network along with an explainable machine learning approach to discriminate between RLS patients and normal controls based on 2D images of cortical current densities. This enabled us to identify the spatial location of critical neural activities on the cortical surface.

## 2. Materials and Methods

### 2.1. Experimental Methods

The details on ERP data were reported previously [5]. ERP signals were recorded from 13 RLS patients and 13 healthy normal controls while performing a modified version of Sternberg’s working memory task. The subjects were cross-matched with the subjects in our previous publication [5]. Subjects were instructed to sleep for at least 8 h on the night before the experiment and asked to refrain from caffeine intake and excessive exercise for at least 24 h prior to the experiment. The task consisted of 200 trials of encoding, retention, and retrieval. During the encoding, two, three, or four single digits between zero and nine were displayed randomly on a screen, after presenting a cue sign (‘+’) for visual orienting. After the retention interval of 2200 ms following the presentation of the digits, a target number between zero and nine was presented, and the subjects should respond as soon as possible by pressing a button if the target item was presented previously during the encoding phase. The subjects were instructed to respond with either their left (matched items) or right hand (unmatched items). Each single trial lasted for 7.8–10.6 s, depending on the number of items (memory load size).

At the end of each trial, feedback for the response was presented as “correct” or “incorrect”. Both reaction time and the correct answer rate (hit rate) were measured. The visual stimuli were presented using commercial software (PRESENTATION; Neurobehavioral Systems, Berkeley, CA, USA).

Electroencephalogram signals were recorded from 19 electrodes using the AS40-PLUS amplifier system (GRASS^®^ Comet-PLUS^®^, West Warwick, RI, USA) in this study. The impedances for all electrode channels were kept below 10 kΩ. The signals were amplified and filtered by a bandpass filter with cut-off frequencies of 0.1–70 Hz and stored with a sampling rate of 400 samples/s.

The single-trial ERP waveforms were segmented from −200 ms to 1500 ms using the probe stimulus during the retrieval phase. The waveforms severely contaminated by non-stereotyped artifacts, such as drift or high-frequency noise, were removed by visual inspection. In addition, the waveforms were excluded if the absolute value of the electrooculogram was larger than 100 µV. Finally, independent component analysis was performed to correct stereotyped artifacts such as ocular and muscular artifacts. All the erroneous trials were eliminated from further analysis.

Before the experiment, all subjects were requested to complete the tests for the assessments of sleep qualities and RLS symptoms. The international RLS Severity Scale (IRLS) score was used to assess the severity of RLS symptoms [22]. Sleep quality and disturbances were assessed using the Pittsburgh Sleep Quality Index (PSQI) [23]. The Epworth Sleepiness Scale (ESS) represents the subject’s general level of daytime sleepiness [24]. An impact of insomnia was evaluated by the Insomnia Severity Index (ISI) [25]. A severity of depressive symptom was estimated using the Korean version of Back Depression Inventory II (BDI) [26]. The Hospital Anxiety and Depression Scale (HADS) was used to quantify the symptoms of anxiety (HADS Anxiety) and depression (HADS depression) [27].

### 2.2. Data Analysis Methods

Convolutional neural networks (CNNs) were applied to cortical current density derived from ERP to classify RLS patients and normal controls [28]. The input features critical to the classification outputs of the CNNs were obtained through layer-wise relevance propagation (LRP) [29]. The entire analysis consisted of the following steps: (1) generating input data through cortical current density estimation from a single trial ERP using Mollweide projection [30], (2) learning 2D CNN using the generated data and assessing the performance of the classifier, and (3) calculating the contribution of each input feature to the determination of the CNN classification by applying LRP to the well-developed CNN.

#### 2.2.1. Preparation of Input Data

The signals from four RLS patients were discarded because of heavy contamination of noise and artifacts. A single-trial ERP waveform was extracted from −200 ms to 1500 ms according to the target onset (retrieval phase) and converted to the current density time-series at 15,002 points on the cortical surface by standardized low-resolution brain electromagnetic tomography (sLORETA) [31]. The boundary element method was applied to solve the forward problem in the Brainstorm toolbox [32]. A critical temporal period was determined to be 150–250 ms, because it is known to be dedicated to major cognitive processing to recall an item in WM [7,33,34]. Thus, a map of a 2D image of cortical current densities averaged during this critical period was obtained as the input to the CNN classifier.

The current densities at 15,002 cortical points were first projected onto a spherical surface and then reconstructed to a rectangular 2D image by applying Mollweide projection, a method to flatten the surface of a sphere onto a 2D plane so that the ratio of the equator to half of meridian is 2:1, which is effective to preserve the sizes of each area on the sphere after flattening [30,35]. The gap between the latitude lines and the horizontal lines on the sphere becomes narrower at high latitudes, in order to preserve the relative sizes of all the areas. The left and right hemispheres were projected onto two square images of 60 × 60 pixels, resulting in a 2D input image with 60 × 120 pixels, representing a single-trial cortical activity. The pixel intensities were transformed to z-scores by standardization.

#### 2.2.2. Convolutional Neural Network Classifier

A pattern classifier based on CNN was devised to classify RLS patients and normal people using the 2D images representing single-trial cortical activities. The CNN is recognized to be efficient for image processing as it can learn the spatial structure of the data by mimicking visual information processing of the human brain [36,37]. The local spatial structure of the data is exploited to extract underlying information based on multiple convolutional filters. Our input data are suitable for the CNN as they consist of the current densities on the cortical surface, which is essentially a 2D image. By applying the methods for the image processing to multichannel EEG, recently, it has been shown that the CNN provides successful applications such as motor imagery classification, seizure detection, and sleep stage scoring [38,39,40].

Here, we adopted the structure of VGGNet, which is known to be optimized for image classification [28]. The VGGNet is different from a typical CNN architecture consisting of the repetitions of a convolutional layer and a pooling layer, in that the number of features can be preserved after passing through multiple layers. This advantage is due to the use of a convolution layer with a fixed size (3 × 3) and the reduced use of pooling layers. Because of the stack of deeper layers while preserving the feature sizes, the classification performance can be improved [28].

Figure 1 illustrates the detailed structure of the CNN classifier. The CNN consisted of two 3 × 3 convolutional layers and one 2 × 2 max pooling layer, repeated twice. Two identical convolutional layers were followed by a single max pooling layer. The outputs from the last pooling layer were 128 2D images of 12 × 27 sizes. All convolutional layers were followed by a rectified linear unit (ReLU) activation function. After being converted to the vectors with 41,472 (=128 × 12 × 27) dimensions, these were connected to three fully connected layers, which yielded the outputs corresponding to the class labels. A 50% dropout rate was applied to the first and second fully connected layers after the ReLU activation function. The output of the last fully connected layer is connected to a sigmoid activation function.

#### 2.2.3. Training and Test of the Classifier

In total, 3594 single-trial cortical activities were obtained (RLS patients: 1580, normal controls: 2014). For each session of training, 1580 trials were randomly selected from the normal controls to eliminate the class imbalance problem [41]. We used the leave-one-subject-out cross validation (LOOCV) 22 times because our subjects include 22 persons (13 normal controls, 9 RLS patients) [41]. Each time, the CNN was trained for 21 subjects, while one subject’s data were used for the test of the trained classifier. For each training session, 1/11 of the training data were used as a validation set for early stopping [42], that is, if the accuracy for the validation set did not change for more than 20 epochs, the training was terminated. The binary cross entropy loss function and Adam optimizer were used for training CNN [43]. Grid search was applied to determine the optimal learning rate for each training session [44]. The algorithm was developed based on an open-source machine learning library, PyTorch, with four NVIDIA GeForce RTX 2080 Ti GPUs.

#### 2.2.4. Determination of Critical Input Features by LRP

We determined the input nodes that were critical to the CNN classifier by LRP [29]. The LRP was applied to all of the trained classifiers, not choosing a specific classifier, then the LRP heatmap was obtained for the correctly classified test data. This enabled finding the cortical regions reflecting the differences in neural activities between the RLS and normal control groups. The LRP is a method to determine the contribution of a single input node to the final output by repetitive decomposition of a single node’s output into the contributions from all of the nodes in the previous layer, in the backward direction [45], as can be illustrated by Equation (1):(1)$$R_{j}=\underset{k}{\sum }\frac{z_{jk}}{\sum_{j}z_{jk}}R_{k}$$

Here, $R_{j}$ is the relevance score, which quantifies the contribution of a single node j, and $z_{jk}$ is the influence of node k in the next layer on node j. $R_{j}$ is calculated from the relevance scores of the next layer $R_{k}$, i.e., in the backward direction. The weight $z_{jk}$ is calculated from the activation of each node and the weights of the trained network. Equation (1) represents the basic propagation rule (LRP0 rule), which redistributes the contribution of the higher layer on the lower layer [45]. The basic rule was varied to obtain a less noisy distribution of relevance scores at input nodes, as shown below in Equations (2) and (3) [45]. By adding a small positive number ε to the denominator of Equation (1), the stability of the LRP result is enhanced so that the relevance score does not diverge, as follows:(2)$$R_{j}=\underset{k}{\sum }\frac{z_{jk}}{\varepsilon +\sum_{j}z_{jk}}R_{k}$$

Equation (2) represents the LRP-epsilon rule. Another variation (Equation (3)) represents the LRP-gamma rule.
(3)$$R_{j}=\underset{k}{\sum }\frac{z_{jk}+\gamma \cdot z_{jk}^{+}}{\sum_{j}\left(z_{jk}+\gamma \cdot z_{jk}^{+}\right)}R_{k}$$

Here, $z_{jk}^{+}$ is the positive part of $z_{jk}$, which indicates a positive contribution to prediction. γ denotes the control parameter. As γ increases, the negative effect for the prediction is reduced We adopted the LRP0 rule for the fully connected layers, the LRP-epsilon rule for the middle layers, and the LRP-gamma rule for the lower layers, as shown in Figure 1.

A heatmap representing the relevance of each cortical points was constructed from the relevance scores for the nodes in the input layer. We developed the LRP codes based on the source codes available at http://heatmapping.org (accessed on 17 September 2021). The relevance scores for the nodes in the input layer provide a heatmap representing the contribution of each cortical point to the classifier output.

#### 2.2.5. Statistical Analysis

Repeated measures analysis of variance was applied to investigate the behavioral response. The between-subject factors included group (two levels: RLS patients, normal controls) and the within-subject factors included memory load size (three levels: item 2, item 3, and item 4).

We calculated the Spearman’s rank correlation coefficients to identify the relationship between the neural activity of the identified critical regions and clinical scores for the nine RLS patients included in analysis. For each critical region, Spearman correlation coefficients were calculated between the current density averaged over the cortical points with the relevance scores of the top 3% and clinical scores.

## 3. Results

### 3.1. Behavioral Responses

Detailed results of behavioral responses were reported in our previous paper [5]. The mean hit rate was 96.42% for RLS patients and 96.27% for normal controls. The hit rate was not significantly different between RLS patients and normal controls. The mean reaction time was 852.68 ms for RLS patients and 657.48 ms for normal controls. The reaction time was significantly different between RLS patients and normal controls (F(1,24) = 9.498, *p* < 0.01).

### 3.2. Classifier Performance

The classification accuracy was 99.32 ± 0.49% and 94.05 ± 3.87% for the training and test, respectively. As shown in Figure 2, the area under the receiver operating characteristic (ROC) curve was 0.93, meaning high classification accuracy. The ratios of true positives and true negatives were both above 90%. We could obtain test accuracies over 80% for all of the subjects, as shown in Figure 2b.

### 3.3. Distribution of Critical Features on the Cortical Surface

The cortical regions critical for the discrimination of the RLS patients were identified by the LRP. The heatmaps in Figure 3 show the distribution of the relevance scores on the cortical surface for correct prediction. The critical areas included the left superior frontal, right-left temporal, left superior parietal, lateral occipital, and left insular regions. These areas are known to play important roles in selective attention enforcement and later visual function in working memory recall [46,47]. The areas with relevance scores of the top 3% are shown in blue in Figure 3. By comparing each panel in Figure 3a,b, the areas showing high relevance scores were consistent across different subjects.

### 3.4. Correlations of the Critical Region’s Activities and Clinical Scores

Significant correlations were found in several critical regions, as shown in Table 1. Left insular was correlated with PSQI, ISI, HADS anxiety, and IRLS. PSQI and IRLS were negatively correlated with the left inferior temporal region. ESS was negatively and strongly correlated with the right superior temporal region.

## 4. Discussion

In this study, we developed a CNN-based classifier for the discrimination of RLS patients and normal controls based on single-trial ERPs recorded during a working memory task. In addition to high classification accuracy, our method revealed the characteristics of cortical activities discriminating RLS patients from normal controls. It is remarkable that these results were obtained from single-trial ERP, which is known to suffer from high inter-trial and inter-subject variation as well as a low signal-to-noise ratio, after strict separation of training/test/validation data according to LOOCV.

The hit rate was not significantly different between groups, while the reaction time was significantly slower for the RLS patients as compared with normal controls. This is interpreted as reflecting the delayed responses due to impaired attention in RLS patients [5]. Our analysis methods based on the CNN classifier and LRP successfully identified the relevant cortical areas that underlie attentional information processing.

The single-trial ERPs are contaminated because of a significant amount of noise and distortion, even when we try our best to minimize and remove noise and artifacts. Moreover, the signal amplitude is very small. In addition, significant intra-class variability exists, i.e., the single-trial ERP waveforms are significantly different among the subjects within the same class. We expected that the superb learning and generalization performance of the deep neural network (i.e., CNN) may cope with this problem so that useful buried information can be utilized under a great amount of inter-person variability, noise, and distortion.

The deterioration of frontal function may underlie working memory dysfunction in RLS patients. The 150–250 ms epoch is known to be devoted to selective attention for working memory retrieval, which is exploited to compare the stored and incoming items. Left inferior temporal, left lateral occipital, and superior parietal areas were included in the critical areas identified by the LRP. These are important parts of the secondary visual association area, which affects top-down modulation along with the frontal area [7,8,47,48,49,50]. The top-down modulation contributes to recall efficiency by allocating attention to incoming visual stimuli. These areas are known to be highly activated by more difficult recall tasks [47,51]. Previously, we showed that the response of RLS patients was delayed as compared with normal controls, presumably owing to inefficient top-down modulation of visual information processing for memory recall [5].

The critical areas in Figure 3a included the insular and superior temporal areas. Previous studies reported that the lesions in these operculum areas are associated with semantic memory deficits [52,53,54]. Considering that semantic memory may contribute to working memory efficiency [51,55,56], the dysfunction of these areas in RLS patients may result in reduced behavioral performance as well as the deterioration of verbal fluency test scores reported in a previous study [3,4].

The activities in several critical regions were found to be correlated with some RLS symptoms. The scores representing sleep deprivation, such as PSQI, ISI, HADS anxiety, and IRLS, were correlated with the activities of left inferior temporal and insular regions. Anxiety is highly associated with sleep deprivation [57]. We found that the ESS was significantly correlated with the neural activity of the right superior temporal region. This is in line with the study of Mu et al., which showed that sleep deprivation affects working memory function [8], owing to the difficulty in verbal rehearsal. We interpret these correlations as suggesting that the differences in neural activities between the patients and normal controls originate from a secondary effect due to sleep dysfunction, rather than the intrinsic pathophysiology of RLS.

The results on spatial locations on cortical surfaces should be interpreted with caution because of the inherent limitation of the accuracy of estimating the current source densities from surface EEG recordings [58,59]. Considering that our EEG recordings include a relatively small number of channels, owing to the difficulty in the clinical environment, further studies using higher-density EEG or magnetoencephalogram may be necessary. In addition, it may be possible to improve the performance by adopting more advanced deep neural network structures.

## 5. Conclusions

In conclusion, we developed a method to reveal the neural mechanism of working memory deficits in RLS patients, based on a CNN pattern classifier applied to single-trial cortical current source density and an explainable machine learning approach. This may lead to a useful and easy-to-use prescreening tool for early discrimination of RLS because of the use of single-trial ERPs. Although further tests with a large number of cohorts may be required, we expect that our method will yield a promising result for a larger database, as it was tested based on leave-one-subject-out cross-validation after strict separation of training/test/validation data. The determined critical areas overlapped with the cortical substrates of working memory, and the neural activities in these areas were correlated with some important clinical scores of RLS. These results suggest that the recognized cognitive dysfunction in RLS patients originates from the impairment of neural activities in relevant cortical regions and may contribute to the development of useful biomarkers for the RLS derived from neural signals.

## Figures and Tables

**Figure 1 sensors-22-07792-f001:**
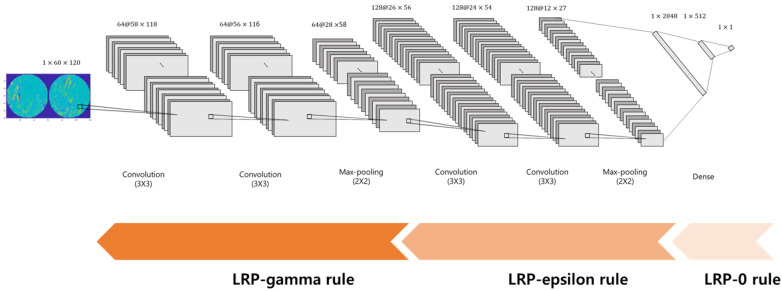
The structure of the CNN classifier for the discrimination of RLS patients and normal controls.

**Figure 2 sensors-22-07792-f002:**
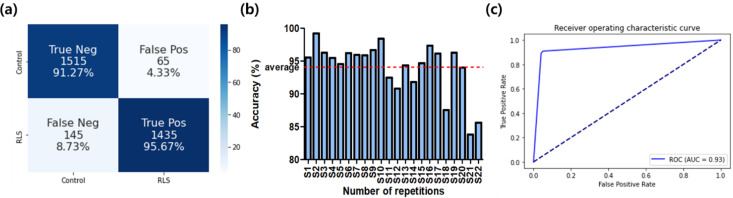
The classification performance of the CNN classifier. (**a**) Confusion matrix; (**b**) classification accuracies for all subjects; (**c**) ROC curve.

**Figure 3 sensors-22-07792-f003:**
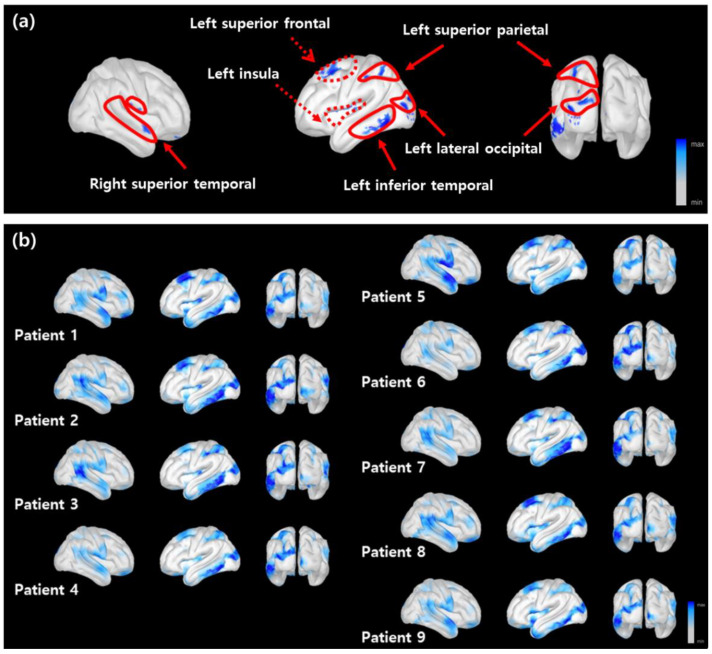
The distribution of the relevance scores on the cortical surface. (**a**) The LRP heatmap averaged over multiple subjects. The critical regions with the top 3% relevance scores (denoted by a blue shading). The solid and dotted lines indicate the cortical areas related to visual processing and executive control, respectively. (**b**) The LRP heatmaps for all subjects obtained from correctly classified results.

**Table 1 sensors-22-07792-t001:** Correlations of critical regions’ activities and clinical scores.

		ESS	ISI	BDI	PSQI	HADS Anxiety	HADS Depression	IRLS
left	Superior frontal	0.185	0.353	0.529	−0.218	0.562	0.458	0.042
	Inferior temporal	0.328	−0.529	−0.361	−0.644 *	−0.468	−0.322	−0.639 *
	Insular	0.227	0.622 *	0.378	0.628 *	0.587 *	0.254	0.630 *
	Superior parietal	−0.210	−0.227	−0.067	0.075	−0.289	−0.068	−0.269
	Lateral occipital	−0.176	0.235	0.429	−0.276	0.196	0.509	0.067
right	Superior temporal	−0.672 *	−0.067	−0.361	−0.243	0.068	−0.322	0.168

*: *p* < 0.05. IRLS, International RLS Severity Scale; ESS, Epworth Sleepiness Scale; ISI, Insomnia Severity Index; BDI-II, Beck Depression Inventory-II; PSQI, Pittsburgh Sleep Quality Index; HADS, Hospital Anxiety and Depression Scale.

## Data Availability

The data are not publicly available owing to privacy issues. The data presented in this study are available upon request from the corresponding author. If necessary, please contact the author.

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
