# Peer review of "Explainable Machine-Learning-Based Characterization of Abnormal Cortical Activities for Working Memory of Restless Legs Syndrome Patients"

_sensors, 2022, doi:10.3390/s22207792_

Round 1

Reviewer 1 Report

The authors investigated the neural mechanism of working memory deficits of restless legs syndrome (RLS) using a CNN-based classifier that discriminated between the RLS patients and normal controls. The novelty of their study is that they explored discriminative cortical regions between the RLS patients and normal controls in the trained model, using LRP. Their approach is interesting and deserves to be published if several concerns are addressed.

Major issues:

1 The behavioral measures should be reported. The authors used Sternberg’s working memory task to evaluate the working memory performance of the participants. Each participant group's task scores (e.g., the correct answer rate and reaction time) should be reported and discussed. Is there a difference between the groups? Is it consistent with the interpretation of the neural data?

2 For the model training process, which model is used for determining critical input features by LRP? You have a number of trained models because used LOOCV. How did you choose the model to be analyzed? What metrics were used?

Does it not affect the LRP results? The stability and reproducibility of the LRP result should be discussed in terms of these concerns.

3 The clinical scores appeared suddenly in the Results section. They should be described in the Materials and Methods section. Due to that, it is hard to understand the contents of Section 3.3. Are the results of Table 1 calculated only for the RLS patients?  It is helpful if such a calculation process are described in Section 2.2.4.

Minor issues:

References 37 and 38 are the same.

Reviewer 2 Report

Title.

It may be possible to reduce the length of the title. 

Introduction

It is only mentioned that the current systems are insufficient, but it is necessary to justify them with data, information and references. The definition of the problem is insufficient and can be extended more.

It is mentioned that the authors try to carry out certain activities. The correct thing to say is what they did, not what they tried. 

What are the disadvantages of the above methods that help you to propose a system based on machine learning? By answering that, you will understand the problem you are looking to solve. Although I am sure you know the problem, you do not explain it adequately. 

Mention a paragraph at the end of this section in which you say how your paper is organized.

Methodology

It is important to mention the brands of electroencephalographs, as they do not all have the same levels of accuracy, and to indicate whether they were properly calibrated.

Please add a blank copy of the patient's informed consent form as supplementary material.

Why was CNN chosen, and what advantages did it offer? Why is there no reference in this subsection 2.2? 

Be sure to define all acronyms.

Why in section 2.2.4 is there no reference? It would be best if you justified why using equation 1.

Results

What part of the objective do the correlations report help to solve?

If correlations are to be reported, they should be stated in the methodology and associated with some objective.

What do correlations that were not statistically significant mean? 

I suggest putting a section on future work. 

Reviewer 3 Report

This paper proposes a deep learning model to identify the neural mechanism of working memory deficits that might be associated with restless legs syndrome (RLS).

The paper is interesting as well as the topic approach. However, I have selected below some points that need more clarification and improvement.

- The text needs to be reviewed for English grammar.

- Abstract: Overall it needs more clarification. It's not clear the main motivation: Why identify the cortical regions? Cant it be done by evaluating the MRI data without any machine learning approach? Also, the authors should include which type of data was used in this work.

- Some references are cited as numbers and others as the authors' initials.

- Before the "Materials and Methods" the authors could include a "Related Works" section to guide the reader to the state of the art in this field.

- Fig 1: The length of the layers is very small.

- Sec. 2.2.2: Although the authors used the VGGNet as a deep neural network, would be great to see their input on other models as well, since VGG is an old model.

- Sec. 3.3: It is not clear how the correlation was calculated.

- The authors mentioned the AUC ROC. Please, include in the paper the plot of this curve.

Reviewer 4 Report

Overall this is a well-written manuscript, I do have some questions for the authors.

Introduction

This is a concise and well written introduction

Methods

1. I understand that you have already published part of this work, but can you specify in this paper whether subjects were cross-matched? (even though you mention that they were age matched in your previous study) Were the controls recruited after the RSL patients?

2. Which EEG system did you use?

3. Approximately how long did the trials last?

I could not access the original publication, so these questions should be answered for other readers who also may not have access to the original publication

1. Did you control for caffeine, prior night's sleep (the night before the test could've been especially bad or especially good), and last 24 hours exercise in your trials? 

You do an excellent job of writing the analyses aspect

Results

While i agree with what you are saying on lines 187-189, please move that to the discussion section

What is the purpose of lines 205-207?

In table 1 you should have a legend 

You have HADS Anxiety and Depression in Table 1, what surveys are these?

Discussion

You bring up the association between the BDI and regions of the brian that are associated with working memory however, the HADS Depression scale does not have any significant correlations. How do you justify that?

What were the limitations of this study?

Round 2

Reviewer 1 Report

Response 2: The LRP was applied to all the trained classifiers, not choosing a specific classifier, and then, the LRP heatmap was obtained for the correctly classified test data. We also found that the LRP results was consistent among multiple subjects, i.e., the identified critical areas was similar for multiple trained CNNs (shown in Figure 2.b). Thus, we judge that the LRP result is stable.

Could you indicate how many subjects were correctly classified? In Figure 3.b, the LRP heatmap was shown for only two subjects. 

The average map (Figure 3.a) is interesting but biased because the LOOCV overlaps its training data (subjects). It would be beneficial if the consistency and stability in LRP maps among multiple subjects were shown by showing and comparing all the "correctly-classified" LRP maps.

Reviewer 2 Report

Thanks for attending to my comments. 

Author Response

Thank you for your comments and suggestions that allowed us to improve the quality of the manuscript.

Reviewer 4 Report

I appreciate the authors addressing my concerns however, the new additions have brought about some other minor concerns

1. Did you try to identify whether data was normally distributed for the RM-ANOVA? You state that you used Spearman's Rho for your correlation, so I am assuming the data was not normally distributed? Did you use any normalization techniques? What happened when you used them?

2. Please specify that you used a Spearman's Rho correlation. 
